# Use of Multivariate Statistics in the Processing of Data on Wine Volatile Compounds Obtained by HS-SPME-GC-MS

**DOI:** 10.3390/foods11070910

**Published:** 2022-03-22

**Authors:** Maria Tufariello, Sandra Pati, Lorenzo Palombi, Francesco Grieco, Ilario Losito

**Affiliations:** 1CNR–Institute of Sciences of Food Production (ISPA), Via Prov. le, Lecce-Monteroni, 73100 Lecce, Italy; maria.tufariello@ispa.cnr.it (M.T.); francesco.grieco@ispa.cnr.it (F.G.); 2Department of Agriculture, Food, Natural Resources and Engineering (DAFNE), University of Foggia, Via Napoli 25, 71100 Foggia, Italy; s.pati@unifg.it; 3CNR—Institute for Applied Physics “Nello Carrara” (IFAC), Via Madonna del Piano 10, Sesto Fiorentino, 50019 Firenze, Italy; 4Department of Chemistry and SMART Inter-Department Research Center, University of Bari “Aldo Moro”, Via E. Orabona 4, 70126 Bari, Italy; ilario.losito@uniba.it

**Keywords:** wine, HS-SPME-GC-MS, volatile compounds, multivariate statistical analysis, artificial intelligence

## Abstract

This review takes a snapshot of the main multivariate statistical techniques and methods used to process data on the concentrations of wine volatile molecules extracted by means of solid phase micro-extraction and analyzed using GC-MS. Hypothesis test, exploratory analysis, regression models, and unsupervised and supervised pattern recognition methods are illustrated and discussed. Several applications in the wine volatolomic sector are described to highlight different interactions among the various matrix components and volatiles. In addition, the use of Artificial Intelligence-based methods is discussed as an innovative class of methods for validating wine varietal authenticity and geographical traceability.

## 1. Introduction

Wine is a very complex matrix composed of molecules of diverse nature and structure (e.g., proteins, amino acids, carbohydrates, phenolic compounds, volatile components, and inorganic compounds), present in a wide range of concentrations [1]. The chemical composition of wine, and its quality as well, is influenced by numerous factors, including grape variety, climate, cultivation practices, geographical location, vintage, yeast strains, and fermentation conditions [1]. The use of metabolomics in the wine sector has provided an effective tool for obtaining a holistic picture of viticulture and winemaking practices that are useful for ensuring the quality and traceability of wine [2,3]. In this scenario, volatolomics play an important role, because the volatile profile constitutes a fingerprint of the aroma properties of a wine, making feasible the identification of adulterations and the traceability of geographical origin of raw materials [2,3,4].

The principal technique used to separate, identify, and quantify volatile organic compounds (VOCs) in wine is Gas Chromatography coupled to Mass Spectrometry (GC-MS) [2]. Wine aroma studies using gas chromatography for the separation and identification of volatiles employ different techniques to achieve accurate and reliable extractability of these analytes. The most common are liquid–liquid extraction, headspace extraction, purge and trap extraction, solid phase extraction (SPE), and solid-phase micro-extraction (SPME). In the last few years, the latter has been extensively used for the study of wine volatiles [5,6,7]. Compared to other techniques, SPME is characterized by simplicity of use, sensitivity, ease of automation, and also does not require the use of solvents and offers the possibility of integrating the two phases of extraction and concentration into a single step. Also, several studies have shown the SPME’s abilities to be versatile and able to cover different aspects in the wine field, such as the origin and flavor characterization [8,9,10]; the off-flavor identification [11,12]; the improvement of the fermentation process [13]; the ageing studies [14,15]; the evaluation of the packaging’s impact on the sensory profile [16]; and the influence of pedo-climatic conditions [17].

The development of the SPME method requires the setup of some fundamental parameters to obtain good extraction efficiency, and good results in terms of accuracy and repeatability. It is necessary to identify the optimal time and temperature conditions for the equilibration and extraction phases, to choose the most suitable fiber according to the analytes to be extracted, to determine the volume of the sample, and the amount of salt to be used. The availability of fibers with different adsorbent polymeric coatings makes SPME a very versatile technique suitable for the extraction of different classes of volatile compounds from different matrices. For specific applications, the choice of a suitable solid phase depends on the class of compounds to be analyzed [18]. Polydimethylsiloxane (PDMS)-coated fibers have shown very good extraction efficiency with non-polar compounds [19], as their coating consists of a non-polar material.

However, in recent years, the use of coatings based on the combination of different adsorbent polymers including poly divinylbenzene (DVB) and carbowax (CAR), such as DVB/CAR/PDMS or CAR/PDMS, has gained popularity, since they can be used for the extraction of a wider range of analytes [20]. The growing interest in this SPME technique, especially focused on the wine headspace (HS), is demonstrated by an almost linear increase in the number of manuscripts related to SPME applications in wine, evidenced from a search on the Scopus database between 1998 and 2021 using the keywords “HS-SPME/wine/volatile”, as shown in Figure 1.

Generally, metabolomic analysis, and, in particular, the volatolomic approach, generates extremely large volumes of data. Traditionally, wine VOCs’ data have been evaluated using a univariate approach, based on assessing the influence of a single variable on the overall aroma. This approach can provide useful information, but not specific indications on the relationships between the variables [21,22]. In order to study complex matrices such as wine, characterized by the presence of different interactions among the various matrix components, multivariate analysis techniques are required. The latter can exploit and determine, beyond the classic covariance between variables, more complex functional relationships that can be used in different ways, depending on the specific research needs. Different goals require different chemometric techniques to be applied, e.g., for data exploration, classification, or curve resolution [23,24]. Principal Component Analysis (PCA) and Discriminant Analysis (DA), in particular, have been extensively applied to characterize wines based on their volatile content [25,26,27,28,29]. Cluster Analysis (CA) has been used to categorize wines based on their volatile composition [30,31].

Analysis Of Variance (ANOVA), along with PCA, CA, and DA, have been used to classify South African wines, according to cultivar based on volatile content [32]. Moreover, the volatile profile, in combination with unsupervised methods, like Hierarchical Clustering Analysis (HCA) and PCA, was exploited for the discrimination or differentiation of grape cultivars and wines [33]. Noble and Ebeler [34] employed multivariate statistics (Generalized Procrustes Analysis, GPA, and Partial Least Squares regression, PLS) in understanding wine flavor, whereas Marengo et al. [35] revealed the differences in the distribution of metals in Nebbiolo-based wines through supervised pattern recognition methods such as Soft Independent Modelling of Class Analogies [36] and Linear Discriminant Analysis [37] with a new interesting approach characterized by use of neural networks [38].

Starting from this background and in the context of a broad evaluation of the literature concerning the quantitative analysis of wine VOCs based on HS-SPME-GC-MS, including the consideration of method development and calibration approaches [39], this review aims to give a brief description of the different multivariate methods employed to process wine VOCs data obtained by HS-SPME-GC-MS, highlighting the main aspects and applications of each of them.

## 2. Processing of HS-SPME-GC-MS Data on Wine Volatiles by Multivariate Statistics

Multivariate statistical methods are used in different ways to process HS-SPME-GC-MS data on wine volatiles. The main approaches are: (i) hypothesis testing, to determine, for example, whether there are statistically significant differences between different datasets obtained under different experimental conditions or treatments [40,41,42,43]; (ii) exploratory analysis, through which the main similarities and dissimilarities between multivariate data are highlighted [4]; (iii) regression models, by which values of the dependent variables are predicted from those of the measured (independent) variables [23,24]; (iv) pattern recognition, aiming to create clusters of similar multivariate data (unsupervised methods/clustering) or to identify relationships between different variables that allow each of the available data to be classified in a class known a priori (supervised methods/classification) [4,44]. Both unsupervised and supervised methods also make it possible to associate an unknown data item with one of the identified clusters or to classify it in one of the a priori known classes. With regard to supervised classification methods, these include a training step, during which a model is optimized for the best possible classification of a multivariate dataset after a prior classification is given for samples. The application of these methods also requires an accurate validation, consisting in evaluating the model performance on further samples, not used for training, whose classification is known a priori. Once validation is completed, the obtained model can be exploited to classify samples whose classification is unknown, starting from multivariate data. In some cases, besides allowing the classification of new samples, it is also possible to study the factors (e.g., the combinations of independent variables) that the method itself identifies as the basis for the best classification.

In some cases, the same multivariate statistical technique can be used for different objectives. For example, analysis of variance techniques (ANOVA-MANOVA) can be used both as hypothesis tests and, by means of various post-hoc methods, to obtain data clustering. Other methods, such as PCA, are used for exploratory analysis, for clustering and to determine latent factors. Other methods, such as PLS or Artificial Neural Networks (ANNs), can be used for both regression and pattern recognition (Figure 2).

The following paragraphs describe the main multivariate statistical methods used for the analysis of HS-SPME-GC-MS data on wine volatiles. For each method, a brief description of its principle of operation, the peculiar characteristics, and the different applications is given.

### 2.1. ANOVA-MANOVA

A great volume of data is usually obtained for wine volatile compounds by HS-SPME-GC-MS analysis, therefore, in order to understand if there are significant differences between the means of data referred to different groups of samples, analysis of variance (ANOVA) is generally the mostly adopted statistical method. ANOVA allows to check if the so called null-hypothesis, i.e., the one stating that samples in all groups are taken from populations with the same mean value, can be accepted or not. If the null hypothesis is rejected, post-hoc methods performing a multiple comparison between the mean values of the groups can be applied to verify which of them differ significantly. Commonly adopted approaches to make multiple comparisons between means under the assumption of Gaussian distribution of data are the Tukey’s and the Tukey–Kramer’s methods, which are appropriate when the within-group variances are comparable with each other and are used for groups including equal or different numbers of data, respectively.

One Way ANOVA described so far is suitable if the effect of a single factor (independent variable) is to be determined; when the effect of more than one factor has to be evaluated, the multi-way ANOVA, also called N-way ANOVA can be used. This technique provides not only information about the effect of a single factor on the dependent variable (peak area or compound concentration), but also about the effects of the interactions among factors. Different contributes are reported in the literature on the application of multi-way ANOVA analysis to determine if there were statistically significant differences between wine varieties [40] and wines aged in wood barrels [41]. Moreno-Olivares et al. [42] used a two-way ANOVA analysis to characterize the volatile profile of new white wine varieties made from *Monastrell* grapes grown in southeastern Spain. The analysis considered different groups of aromatic compounds in wines as a function of variety and vintage, revealing a great variability for different samples in terms of the concentrations obtained in the different families of volatile compounds.

The comparison of multivariate sample means can be obtained by Multivariate ANOVA (MANOVA), in which covariance between dependent variables is used to test the statistical significance of the differences. Pérez-Prieto et al. [41] used MANOVA to study the effects of oak origin, barrel volume, and barrel age on all the constituents measured in wines. Aragoni et al. [43] determined the effects of variety, clarification, temperature, and yeast type on all constituents of the acid fraction, higher alcohols, and conventional parameters of wines obtained from grape must of *Muscatel* and *Malvasia* varieties. MANOVA analysis was employed to compare the variations of volatile compounds in the white wine *Muscat Ottonel* variety aged in the presence of untoasted oak chips, toasted oak chips, and untoasted barrel, considering three ageing periods [45]. ANOVA has also been recruited in descriptive sensory analysis, to check overall differences among the products for aroma, taste, and mouthfeel terms [46].

### 2.2. Principal Component Analysis

Principal Component Analysis (PCA) is a multivariate data statistical analysis that provides a reduction in data set dimensionality by finding linear combinations of the original independent variables, called principal components, which explain the maximum of data-set variance. Principal components are orthogonal latent variables generated from the correlation or covariance matrices of data; original data projections along the axes identified by principal components are called scores. The full set of principal components can be as large as the original set of variables, but most of the original data variance is typically concentrated in a limited number of principal components. Similarities or differences between the original multivariate data can thus be usually appreciated on a simple two-dimensional scatterplot, called a score plot, reporting the scores based on the first two principal components. Moreover, by plotting the contribution of the original variables to the principal components (loading plots) it is also possible to understand how the original variables contribute to the similarity or to the difference between original samples. PCA is used as a tool for screening, extracting, and compressing multivariate data [24,26,27,28,29]. In the context of the HS-SPME-GC-MS analysis of wines’ volatile compounds, PCA is one of the most useful multivariate techniques to assess the authenticity of wines [47,48]. As an example, it was adopted to distinguish 22 red wines produced in the four main wine regions in France, starting from data obtained from both sensory and VOCs analyses [49]. Recently, the combined use of the ANOVA technique to select the most “class-distinguishing” chromatographic peaks area and the subsequent PCA analysis allowed Sudol et al. [50] to cluster white wines (*Grillo* wines) produced in different areas of Sicily and to determine geographic differences in their volatiloma.

Vilanova et al. [51] investigated the correlation between the volatile composition and sensory properties in Spanish *Albariño* wines, through PCA. Following this, the multivariate regression approach based on the use of PLS and PCA was used by Poggesi et al. [52] to study the correlation between sensory data and volatile compounds, in Pinot Blanc, in order to use chemical fingerprints to obtain a prediction of the sensory profile of the wine. PCA analysis was also used to evaluate the impact of different yeast strains on the wine quality and on the progress of the fermentation process. As an example, Tufariello and co-workers [53] applied PCA to identify the volatile compounds that best discriminated wines produced by yeast strains selected in the two different areas, i.e., north and south Salento in the Apulia region of Italy. In addition, PCA was used for highlighting the differences among sparkling wines produced using different autochthonous yeast strains for the secondary fermentation step [54]. PCA coupled with discriminant analysis (DA) analysis has recently been used as a chemometric tool to identify the ageing process (barrel- or chip-aged) the wine undergoes, by selecting key volatile molecules detected via GC-MS [55]. Through a PCA-based elaboration of data on phenolics and volatile compounds, Casassa et al. [56] highlighted that there were significant differences between wines aged in control and new barrels, while fewer clear-cut differences were detected between wines aged in barrels produced with different bending/toasting protocols. More recently, PCA was applied, in conjunction with Hierarchical Cluster Analysis, to a set of 103 volatile compounds identified by HS-SPME-GC-MS, indicating peculiar features in the VOC profile related to the geographical origin of nine red wines produced in Brazil [57]. Maioli et al. [58] used PCA to assess the effect of different tank materials on the profile of VOCs of a *Sangiovese* red wine obtained using HS-SPME-GC-MS. In a further recent study, PCA was adopted to investigate the influence of iron deficiency in the vineyard on the profile of VOCs related to floral notes or green-herbaceous aroma, recognized by HS-SPME-GC-MS in the headspace of wines produced in the *Ribera del Duero* region in North-Central Spain [59].

### 2.3. Hierarchical Cluster Analysis

Cluster Analysis represents a set of unsupervised methods that aim at grouping different samples based on the similarity assessed from a set of multivariate data provided for them. One of the most common methods for Cluster Analysis is *Hierarchical Cluster Analysis* (HCA), which allows a grouping of data without a prior knowledge of the number of clusters to be formed. In particular, the agglomerative approach is usually adopted for HCA, i.e., single samples are progressively grouped in clusters of increasing dimensions based on their distance in the multivariate space, with Euclidean distance being the most adopted. Different agglomerative algorithms, starting from these distances, can be used, like those referred to as “*single*” (shortest distance), “*complete*” (farthest distance) and “*average*” (unweighted average distance) linkage and Ward’s method, based on the minimization of variance inside groups.

Following the metrics and linkage criteria indicated by the user, HCA allows the building of a complete clustering dendrogram, through which a qualitative visualization of grouping among samples in a two-dimensional space is possible. This aspect has made HCA the preferred technique when Cluster Analysis is performed for exploratory purposes in the oenological field. Among reported applications, Marengo et al. [35] used HCA based on Euclidean distances and Ward’s method of agglomeration to evaluate similarities between wine samples produced from the *Nebbiolo* grape in the Langhe and Roero areas (Piedmont, Northern Italy) but differing in vintage (respectively, 3 years, 2 years, 1 year, 8 months and a few months) and production zone, starting from data on volatile compounds. Dall’Asta et al. [60] demonstrated the possibility of classifying high quality wines according to their brand based on their volatile fingerprint using PCA and HCA analysis. HCA was exploited, in synergy with PCA, to study grape and wine aroma [61] i.e., as a tool to find the key aromatic series of pulp juice, skin, and whole berries. This type of investigation is also important in the context of fraud prevention in the oenological field. In research about the authenticity of red wines from Poland, Stoij et al. [62] used HCA analysis to assess that the Polish wines were separated thoroughly from wines produced in other European countries, notably France, Italy, and Spain, starting from data on ethyl phenylacetate, hexan-1-ol, ethyl 2-hydroxy-4-methylpentanoate, (E)-3-hexen-1-ol, 2-phenylethanol and 3-(methylsulfanyl)propan-1-ol. Recently, Valentin et al. [63] identified the chemical profile (including the volatile profile) that characterizes Chilean *Carmérère* wines by using HCA and PCA, starting to establish a database for further analysis of the authenticity of South American wines. Costa et al. [64] used HCA analysis coupled with other statistical techniques to evaluate, for the first time, the impact of mannoproteins on the aroma quality of sparkling wines produced with the *Champenoise* method.

### 2.4. Linear Discriminant Analysis

Linear Discriminant Analysis (LDA) is one of the most used methods to perform supervised pattern recognition. This method is usually based on identifying linear combinations of independent variables (called Linear Discriminant Functions, LDF) maximizing the between-class variance and minimizing the within-class variance. In particular, if k is the number of classes, and if the number of independent variables is larger than k, then k-1 LDFs are identified. The resulting LDF may be used as a linear classifier, or for dimensionality reduction before later classification. LDF can be used as a dimensionality reduction technique as it determines a hyperspace with less dimensions than the original data, on which these can be projected to achieve the best possible linear separation among the given classes. The LDA is based on fairly strong assumptions, i.e., that the classes are linearly separable in the multidimensional space of the independent variables and that the variance–covariance matrices are equal for each class. These assumptions do not allow this type of analysis to be applied without a careful validation. A further condition for the application of the LDA is that the number of samples in the training dataset must be greater than the number of independent variables. If the number of independent variables is greater, “feature selection” techniques, i.e., determining a limited but representative subset of independent variables, or “feature reduction” techniques, that enable a reduction of the size of data without significant loss of information, can be adopted. As previously mentioned, the PCA can be used for this purpose.

Turning to applications of LDA to wine volatiles’ data, the combination of LDA and PCA analysis was successfully applied for the varietal differentiation of *Loureira*, *Dona Branca*, and *Treixadura* wines [28], starting from their volatile profiles. Thirty-four *Sauvignon Blanc* wine samples from three different countries and six regions were analyzed by HS-SPME-GC-MS by Berna et al. [65] and LDA was applied to the resulting data, showing three distinct clusters or classes of wines with different aroma profiles. In particular, wines from the Loire Region in France and wines from Tasmania and Western Australia were found to have similar aroma patterns. In a research on the effect of time and storage conditions on major volatile compounds of *Zalema* white wine, LDA was exploited to distinguish among wines with different conditions and times of storage [66]. Ubeda et al. [67] adopted the synergy between PCA and LDA to find the differences in the volatile profiles among Chilean sparkling wines obtained with different production methods. In addition, LDA, coupled with PCA and multilayer perceptrons’ neural networks (MLP-NN) were used as chemometric tools to differentiate several Spanish white wines according to their geographical origin, using selected volatile compounds as input variables [68]. Similarly, linear discriminant analysis (LDA) was successfully applied to obtain an appropriate classification of white and red wines of various geographical origins using the volatolomic approach [8].

Recently, Moreno-Olivares et al. [42] used LDA to study the volatile profile of different crosses of white wines obtained from *Monastrell* and other varieties, showing that the white crosses obtained from red varieties were aromatically more similar to the white wine than to the respective parental.

### 2.5. Partial Least Squares

If the number of independent variables is greater than the number of samples, it is still possible to use multivariate statistical techniques, such as Partial Least Square (PLS), that are not subject to this constraint. The PLS approach is particularly suited for data containing correlated independent variables, since it constructs new predictor variables, called latent variables or components, as linear combinations of the original ones. PLS is designed to evaluate these components while considering the observed response values. The working principle of PLS is to find a finite number of linear combinations of the independent variables describing its variance as much as possible and at the same time having the largest correlation with the dependent variables. This is obtained by algorithms capable of maximizing the covariance between independent and dependent variables.

Guillén et al. [69], illustrated a study on the possibility of obtaining regression models by means of Partial Least Squares (PLS) and Multiple Linear Regression (MLR) to correlate a series of parameters, such as the concentration of short-chain organic acids, higher alcohols, and phenolic compounds, to the age of vintage *Sherry* wines. PLS was also successfully applied in a study focused on the correlations between volatile compounds of *Albariño* wines and sensory descriptors [51]. As demonstrated by other authors, PLS regression is, among multivariate techniques, the best approach to highlight the correlations between chemical data, obtained by HS-SPME-GC-MS, and sensory descriptors [70,71,72].

If the independent variable is categorical, the PLS technique can be used as a supervised pattern recognition technique. In this case, reference is made to the Partial Least Square Discriminant Analysis (PLS-DA) variant. In this variant, the operating principle remains unchanged, and the dependent categorical variable (the class) is replaced by a suitable “dummy” multivariate variable. This consists of as many single variables as classes. The value of dummy variables is set to 1 if the sample belongs to its corresponding class, and to 0 otherwise. PLS-DA is a compromise between the usual discriminant analysis and a discriminant analysis based on the principal components of the predictor variables. PLS-DA can provide a good insight into the factors leading to effective discrimination between samples by the analysis of the components (the loading vectors) and their related sample scores, which gives it a relevant role in exploratory data analysis. In order to discriminate between selected wines with different geographical origin (Azores, Canary and Madeira Islands) and of different types (white wine, red wine and fortified wine), the volatile profiles were characterized by Perestrelo et al. [73]. The authors applied the PLS-DA to the dataset to obtain a predictive model for classification of examined wine samples according to their geographical origin and type. This information is crucial to prevent fraud and, therefore, to guarantee wine authenticity. In a study on the volatiles of *Chardonnay* wine, PLS-DA was adopted to find the key volatile metabolites able to discriminate wines fermented by different yeast strains [74].

Recently, Oliveira et al. [75] used PLS-DA to successfully study the discrimination in the volatile composition of a 48-month old bottle-stored white wine closed with either cork, micro-agglomerated or synthetic stoppers, revealing the most discriminating volatiles. PLS-DA was also applied, along with PCA, by Licen et al. [10] to discriminate white wines produced in the *Friulano Collio* area in the region of *Friuli* (North-East of Italy) from those produced in other areas of the same region. In a paper by Karabagias et al. [76], PLS-DA based on VOCs recognized by HS-SPME-GC-MS analysis was adopted to assess the differences between dry and semi-sec white wines produced from eight different Greek grape varieties.

### 2.6. Artificial Neural Networks

A further family of techniques for the statistical analysis of multivariate data are the Artificial Neural Networks (ANNs). ANNs are one of the most flexible and performing techniques within the Machine Learning (ML) paradigm. The name of this type of techniques derives from the structure of the algorithm itself, which was originally designed to imitate the learning and operational model of neurons within the brain. ANNs consist of a complex structure of interconnected units that are called artificial neurons. The most widespread neural network models considered in the framework of wine data analysis are the feed-forward ANNs. In this case, single neurons are organized and structured in different layers: an input layer, one or more hidden layers, and an output layer. Typically, each neuron of each intermediate layer is connected with each neuron of the previous layer and each neuron of the next layer. This kind of ANN is called a fully connected network. The individual neurons of each layer operate in parallel and in a very simple way through a function, called transfer function, which the neuron applies to a linear combination of the outputs from the neurons of the previous layer. The weights of the linear combinations determine the functioning of the network itself and are the free parameters that are optimized during the training phase. For neurons belonging to the hidden layers, the transfer function is generally non-linear, typically a sigmoidal function. This type of transfer function allows the network to “learn” highly non-linear relationships between the independent input variables and the dependent output variables and even the use of a mix of continuous and categorical independent/input variables. Conversely, the transfer function of the output layer is chosen according to the type of output desired. The dependent/output variables can be continuous (in this case the network can be trained to perform a regression) or categorical (in this case the network can be trained to perform a classification). The design of an ANN depends on several parameters, called hyper-parameters, which must be carefully chosen. Among these, the main ones are the various transfer functions in the different layers, the number of hidden layers and the number of neurons that constitute them and various parameters linked to the specific optimization algorithm used. Generally, the regression or pattern recognition models constituted by neural networks have a high number of free parameters to optimize. This aspect necessarily requires a number of available samples greater than the number of independent variables used. To avoid overfitting phenomena, it is also necessary to carry out an accurate model validation. This validation can be carried out directly during the training phase by imposing an early stop when performance on a validation dataset tends to deteriorate.

Over the last 20 years, ANNs have found application in the wine studies for various purposes, including authenticity and traceability assessment [38,77,78], discrimination between treatments [79], and wines [80]. As regards the use of VOCs’ data as an input for training the networks, 35 volatile compounds were used by Marengo et al. [35] as input to a Self-Organizing Map to obtain clusters related to wines’ varietal origin and vintage. Kruzlicova and co-workers [81] demonstrated the possibility to use them for the classification of white varietal wines. In particular, they employed ANNs to classify Slovak white wines of different variety, year of production and from different producers by using, as independent variables, volatile species analyzed by the GC–MS technique.

Jurado et al. [68] employed MultiLayer Perceptrons Neural Networks (MLP-NN), together with PCA and LDA, as chemometrics tools to differentiate Spanish white wines according to their geographical origin. In particular, they highlighted the possibility to identify the product’s geographical origin and authenticity, using the volatile compounds and the chemical composition as input data. Recently, some authors [82] have illustrated the use of the machine learning modelling strategies, using weather and water management information from a *Pinot noir* vineyard from 2008 to 2016 vintages as inputs and aroma profiles from wines from the same vintages, assessed using gas chromatography and chemometrics analyses, as targets. The results showed that the ANN models produced a high accuracy in the prediction of aroma profiles (Model 1; R = 0.99) and chemometrics wine parameters (Model 2; R = 0.94) with no indication of overfitting. These models could offer powerful tools to winemakers to assess the aroma profiles of wines before winemaking, which could help them to adjust some techniques to maintain/increase the quality of wines or wine styles that are characteristic of specific vineyards or regions. These models can be modified for different cultivars and regions by including more data from vertical vintages, to implement artificial intelligence in winemaking. However, the use of ANNs in wine volatolomics remains little exploited compared with the other chemometrics’ techniques. The prediction of wine process parameters is an ambitious objective, as fermentation is a very dynamic process that depends on many variables [83]. In order to address this type of problem, several ANN architectures are available specifically focused on the study of dynamic processes. These are, however, more difficult to implement and require a considerable amount of diversified data, that make them more suitable for training and use in industrial scale plants, than for laboratory experiments [84].

The limited use of these instruments is also thought to be due to a greater difficulty in interpreting the results obtained. The ANNs are in fact generally considered as “black boxes” that give very good results, but without giving explanations on how they obtain them. This factor considerably limits the possibilities to provide interpretations and discussions of the results [85]. However, it should be noted that lately this problem has become an important focus of AI (Artificial Intelligence) research that is starting to provide valid and consolidated tools for the interpretation of the cause–effect links exploited by these types of algorithms to obtain the required results [86].

## 3. Conclusions

The large volume of data generally provided by the HS-SPME-GC-MS analysis of wine volatile compounds represents a precious repository of information on wines and multivariate analysis techniques are a powerful tool to recover the highest possible amount of that information. As described in the present review, the awareness of such potential is constantly increasing in the oenological context and several chemometrics techniques, including PCA, LDA, PLS and ANN, have been applied successfully to evaluate different aspects, like the relationship between viticulture and winemaking practices and the wine aroma profile or the recognition of authenticity or geographical origin of the product. Further research is still needed to enhance the use of ANNs, too little exploited, likely due both to the requirement of a large amount of data and the difficulty to interpret the obtained results. This overview of the chemometrics’ techniques and their application constraints, together with the description of their specific applications in wine volatile HS-SPME-GC-MS studies, could help to increase the awareness of such potential and improve research advances in wine volatolomics. Table 1 gives an overview of the statistical techniques described. For each of them, the general scope, pros and cons (with particular reference to the criticalities introduced by the pre-processing of the data and the size of the available dataset) are described. The main applications mentioned in this review are also summarized.

## Figures and Tables

**Figure 1 foods-11-00910-f001:**
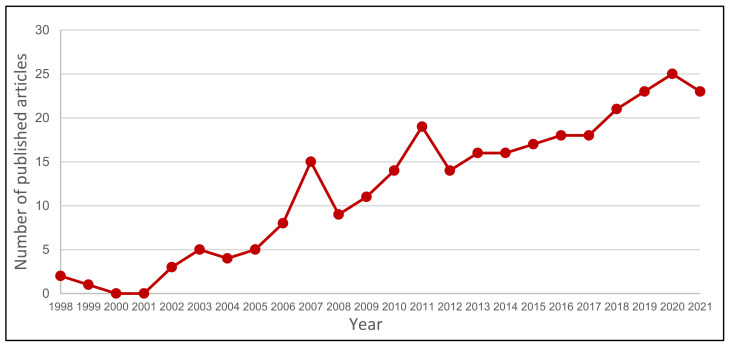
Number of published articles between 1998 and 2022 related to HS-SPME/wine/volatile.

**Figure 2 foods-11-00910-f002:**
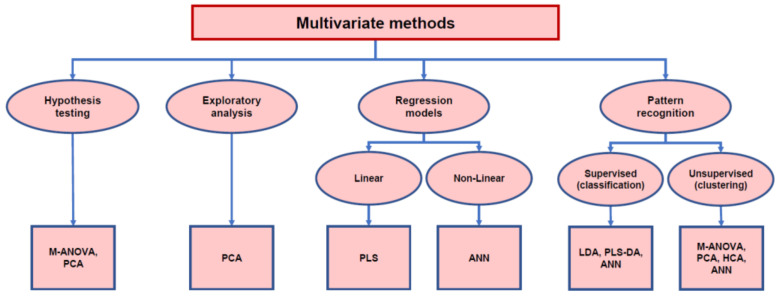
Schematic diagram of different objectives and multivariate statistical analysis techniques used for HS-SPME/GC-MS data.

**Table 1 foods-11-00910-t001:** Overview of the statistical techniques, pro, cons and applications to HS-SPME/GC-MS data.

Name	Scope	Pros	Cons	Applications
M-ANOVA	Hypothesis testing	M-ANOVA allows a deeper analysis than ANOVA in determining changes introduced by a given factor.	It requires a larger number of samples than the number of variables. The extension of the analysis to N factors is more complex. Results can be misleading if the working assumptions are not respected.	Determination of significant differences between wine varieties [40]; characterization of the volatile profile [42]; effects of oak origin, barrel volume, and barrel age [41]; effects of variety, clarification, temperature, and yeast type [43]; descriptive sensory analysis [46].
PCA	Hypothesis testing; Exploratory analysis; Unsupervised classification	Explain multivariate variance by a limited number of factors. It does not suffer the possible multi-collinearity between variables; on the contrary it exploits it. It allows to visualize both the similarity and dissimilarity between samples and the correlation and influence of variables.	Highly dependent on the pre-treatment of the data, e.g., standardization. Sensitive to outliers. The detection of orthogonal (uncorrelated) factors can lead to a misinterpretation of the true cause-effect relationship. Only Euclidean metrics can be considered.	Assessment of the authenticity of wines [47,48]; distinguishing different wines [49]; correlation between volatile composition and sensory properties [51,52]; discrimination of wines produced by selected yeast strains [53]; identification of key-role molecules in aging process [55]; identification of peculiar features in the VOC profile [57].
PLS	Linear regression	It can be used in cases where the number of variables is greater than the number of samples. Handles well any multi-collinearity between variables.	The interpretation of the results is more complex than that of the results of a simple multilinear regression. Results can be poor in the case of non-linear relationships between variables.	Correlation between VOCs and wine ageing [69]; correlations between volatile compounds and sensory descriptors [51]; correlations between chemical data and sensory descriptors [70,71,72].
ANN	Non-linear regression; Supervised classification; Unsupervised classification	Capable to handle strong non-linearity in the underlying model. They are robust to the presence of noise and outliers. They are unaffected by, and indeed exploit, multi-collinearity between variables.	A large number of samples is required. Interpretability of results is more difficult. Validation of results is necessary to exclude overfitting.	Authenticity and traceability assessment [38,77,78]; discrimination between treatments [79] and wines [80]; clustering of wines by varietal origin and vintage; high accuracy in the prediction of aroma profiles from weather and water management information [82]; prediction of wine process parameters [83,84].
LDA	Supervised classification	Interpretability of results is straightforward.	It cannot be used if the number of variables exceeds the number of samples. Conditioned by multi-collinearity. Results can be poor in the case groups are non-linearly separable.	Varietal differentiation from volatile profiles [28]; classification of wines with different aroma profiles [65]; distinguishing among wines with different conditions and times of storage [66].
PLS-DA	Supervised classification	It can be used in cases where the number of variables is greater than the number of samples. Handles well any multi-collinearity between variables	The interpretation of the results is more complex than that of the results of a simple LDA.	Discrimination of selected wines with different geographical origin and type [73]; identification of key volatile metabolites able to discriminate different wines by origin, fermentation, type [74,75,76].
HCA	Unsupervised classification	Straightforward interpretation. It allows different levels of clustering to be evaluated. It allows the use of metrics other than Euclidean to assess similarity and dissimilarity between samples.	The results are highly dependent on the pre-treatment of the data, e.g., whether or not standardization is applied.	Classification of high-quality wines according to their brand based on their volatile fingerprint [60]; fraud prevention by verification of authenticity of wines [62,63,64].

## Data Availability

Data is contained within the article.

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
