# Peer review of "Use of Multivariate Statistics in the Processing of Data on Wine Volatile Compounds Obtained by HS-SPME-GC-MS"

_foods, 2022, doi:10.3390/foods11070910_

Round 1
Reviewer 1 Report
The manuscript “Use of multivariate statistics in the processing of data on wine volatile compounds obtained by HS-SPME-GC-MS” provides a review of the main multivariate statistical techniques used to interpret the data collected from HS-SPME-GC-MS analysis of wines. The article is well written and easy to understand, but I think that it could be enriched with a table summarising all the statistical methods referred in the article, including main advantages and constraints, and some examples of applications in wine research with references. I also think that Figure 1 should have axis titles (Year in horizontal x axis and Number of published articles in vertical y axis).
Author Response
We thank the reviewer for their comments. We agree with the reviewer on the usefulness of a table summarising the uses, advantages and disadvantages of the statistical methods described. To this end, a table was added to the conclusions, describing for each method: the general purposes, pros and cons (including considerations on the number of samples needed in relation to the number of variables considered) and some examples of their application.
We also agree that titles should be added to the axes in Figure 1. Titles have been added.
Reviewer 2 Report
Recently, the analysis of the wine volatile composition has become relevant, especially due to the trend to produce unique and different wines. In this sense, the review provides an interesting summary of the statistical techniques available to understand in greater depth the effect of different variables on the wine sensory profile.
The review is well written and organized. I have no comments on the embedded information or the way it is described.
I would like to ask to the authors the following:
How can the number of data affect the choice of one technique or another? I think it is very interesting to incorporate information related to the number of data (or extension of the database) and the pre-treatment required previous the statistical technique.
I also consider, that a more critical table or section of the techniques described should be incorporated. It can be a table indicating which technique is recommended, for example, to identify, compare, evaluate the impact of some factor, etc. While some of the techniques described can be used for all purposes, not all of them are for everything.
Also, a table or subsection of advantages and disadvantages of the described techniques is desirable.
Author Response
We thank the reviewer for their comments. We agree with the reviewer on the usefulness of a summary table of the various statistical methods described in the review. To this end, a table was added to the conclusions describing for each method: the general aims, pros and cons (including considerations on the number of samples needed in relation to the number of variables, as well as robustness to the presence of outliers, effects of data preprocessing, multicollinearity between variables) and some examples of their application.
Reviewer 3 Report
The review paper explains different multivariate techniques that could be used for data obtained by SPME-GC/MS analysis.
The paper is well-written and clear but the need for this kind of paper is questionable. The main complaint is that these multivariate techniques are uniform for any data, there is no special version that is more suitable for data obtained by SPME-GC/MS.
I suggest you to rewrite the title and abstract, in order to remove accent on SPME-GC/MS (now it is misleading).
Additionally, the English should be uniform- some word are written in American English, and some in British English.
Author Response
We thank the reviewer for their comments.
We agree with the reviewer's observation that the statistical techniques discussed in our manuscript are applicable to any data and not only to data obtained by HS-SPME-GC-MS. It is not the intention of the authors to imply that the statistical techniques described are specific to chemometric statistical analysis, nor specific to the analysis of volatolomic data, nor specific to data obtained by a specific measurement technique. In the light of this, as there are many applications of the multivariate approach, we wanted to guide the reader's attention towards the use of the different chemometric techniques in the field of wine volatolomics and, in particular, on the key role of these techniques on the statistical analysis of volatile molecules data, a subject already discussed in the review published in Processes. Having already discussed in a previous manuscript the critical aspects related to the different methods of quantification of volatiles extracted by the HS-SPME technique, we wanted to deepen the chemometric applications in the same field of research. It is the opinion of the authors that any change of title could be misleading as for all intents and purposes the review focused on applications to data obtained through HS-SPME-GC-MS.
English has been checked and uniformized to British English.
Round 2
Reviewer 2 Report
No comments
Reviewer 3 Report
The authors' explanations are accepted and I suggest editor to accept the revised manuscript.